# Description and Use of Three-Dimensional Numerical Phantoms of Cardiac Computed Tomography Images

Miguel Vera [1,*] , Antonio Bravo [1] and Rubén Medina [2]

1 Facultad de Ciencias Básicas y Biomédicas, Universidad Simón Bolívar, Cúcuta 540004, Colombia
2 CIBYTEL-Engineering School, Universidad de Los Andes, Núcleo La Hechicera, Mérida 5101, Venezuela
* Correspondence: m.avera@unisimonbolivar.edu.co

**Abstract:** The World Health Organization indicates the top cause of death is heart disease. These diseases can be detected using several imaging modalities, especially cardiac computed tomography (CT), whose images have imperfections associated with noise and certain artifacts. To minimize the impact of these imperfections on the quality of the CT images, several researchers have developed digital image processing techniques (DPIT) by which the quality is evaluated considering several metrics and databases (DB), both real and simulated. This article describes the processes that made it possible to generate and utilize six three-dimensional synthetic cardiac DBs or voxels-based numerical phantoms. An exhaustive analysis of the most relevant features of images of the left ventricle, belonging to a real CT DB of the human heart, was performed. These features are recreated in the synthetic DBs, generating a reference phantom or ground truth free of imperfections (DB1) and five phantoms, in which Poisson noise (DB2), stair-step artifact (DB3), streak artifact (DB4), both artifacts (DB5) and all imperfections (DB6) are incorporated. These DBs can be used to determine the performance of DPIT, aimed at decreasing the effect of these imperfections on the quality of cardiac images.

**Keywords:** numerical phantoms; cardiac dataset; processing techniques; artifacts; Poisson noise





## 1. Summary

In spite of the evolution experienced in cardiac imaging, based on X-ray emission, with multiple slices, which is known as computed tomography (CT), a standard model has not yet been achieved due to situations related to the presence of imperfections or problems, linked to defects caused by various artifacts and the noise generated during the image acquisition process in this modality [1,2]. Although there are several sub-modalities within computed tomography, the differences are not relevant. In this article, CT is considered the modality that genuinely represents the main characteristics of computed images generated by X-ray emission.

Therefore, it is necessary to point out that the most important artifacts present in real cardiac CT databases, synchronized by electrocardiographic signal, are those of type staircase and dark band [3,4]. The first artifact occurs when the electrocardiogram trigger is not correctly synchronized with the cardiac phase or when it overlaps the reconstructed sections. In contrast, the second is caused when X-rays pass through structures containing bone or contrast media [5–7].

Figure 1 shows the streak and stair-step artifacts in two real cardiac CT databases.

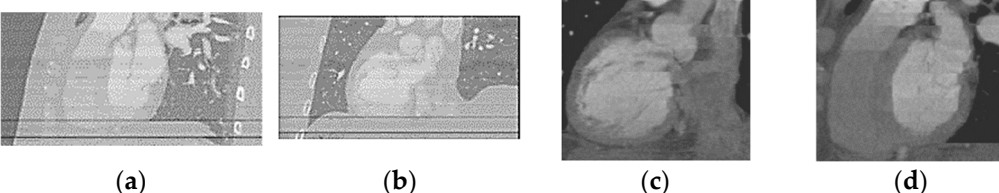

|        |        |        |        |
| :----: | :----: | :----: | :----: |
| (**a**) | (**b**) | (**c**) | (**d**) |

**Figure 1.** Artifacts in real CT images: (**a,b**) streak artifact in sagittal and coronal views. (**c,d**) Stair-step artifact in coronal and sagittal views.

The literature reports that the dominant noise in computed tomography is the Poisson type [8]. Generally, this type of noise is directly proportional to the intensity or gray level of a voxel, implying that voxels of higher intensity are more affected than those of lower intensity in images contaminated by this imperfection [9–11]. The mathematical model governing Poisson noise is:

$$p(X = k) = \frac{\lambda^k e^{-\lambda}}{k!} \tag{1}$$

where $p$ defines a discrete probability distribution, with $k \in \{0, 1, 2, \ldots\}$ representing the number of times a probable event occurs; the parameter $\lambda \in (0, \infty)$ represents the number of times a phenomenon is expected to occur; while $X$ is the random variable associated with $p$.

This distribution has been sufficiently characterized in the literature, and its main limitation is that it is only valid for "small" $k$ values. This aspect is analyzed in Section 3.1 of this article. In Figure 2, axial views of the Poisson noise are shown considering a real cardiac CT database.

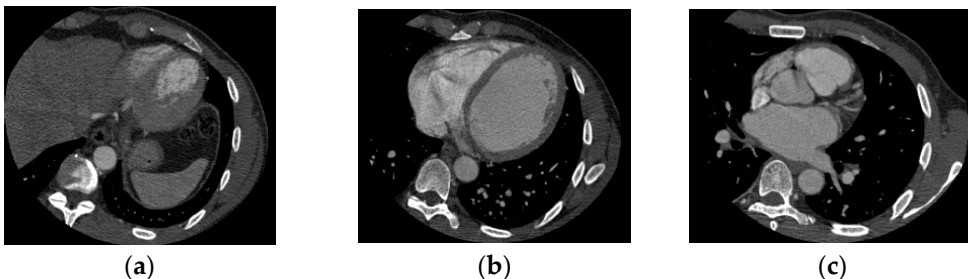

|        |        |        |
| :----: | :----: | :----: |
| (**a**) | (**b**) | (**c**) |

**Figure 2.** Evidence of Poisson noise in real cardiac CT database: (**a**) left slice. (**b**) Center slice. (**c**) Right slice.

Of all the types of phantoms that exist (physical, realistic, numerical, geometrical and commercial phantoms, among others), in the present article, the interest is focused on numerical phantoms [12]. In this sense, we describe the creation of a dataset consisting of six synthetic databases or phantoms of this type based on densitometric information of the images represented in the three-dimensional domain [13]. In them, the most frequent imperfections in cardiac computed tomography, corresponding to Poisson noise and both dark band and staircase artifacts, have been re-created [14]. These databases were created during the development of the doctoral thesis referenced in Ref. [15].

A review of both the literature and publicly available database repositories reveals an interesting proliferation of phantoms in various application areas. Therefore, by means of Table 1, a summary of synthetic databases linked to the human body is presented.

Based on an analysis of the information reported in Table 1, for the computed tomography modality, it is evident that no database was found that exhibited similar or comparable characteristics to the dataset addressed in this article. In that search, only one anthropomorphic-type phantom of the left ventricle was detected (see Ref. [27]), which was not included in Table 1 because it was not numerical. For these reasons, it can be stated that there is no numerical synthetic benchmark dataset available that would allow the development of preliminary experiments to assess the quality of digital image processing

techniques before tackling the real data, and this fact gives important meaning to the originality of the synthetic cardiac dataset presented in this article.

**Table 1.** A brief review of the state of the art regarding numerical phantoms and their purpose.

| References | Application Area (Considered Image Modality) |
|---|---|
| Li et al. [16] | Mitral valve reparation (CT-US [a]) |
| Müller et al. [17] | Human breathing model for detecting irregularities (CT) |
| Lyu et al. [18] | Liver segmentation (CT) |
| Nomura et al. [19] | Streak artifact assessment (Cone Beam-CT) |
| Dossing et al. [20] | Identification of crystals embedded in a phantom (Dual Energy-CT) |
| Medici et al. [21] | Effect of the geometry of 4 phantoms in the detection of the minimum detectable radionuclide activity (Gammagraphy) |
| Lubis et al. [22] | Establishing an adequate correspondence between images of classic angiographic computed tomography and rotational angiography (CT) |
| Pezhman et al. [23] | Determining factors involved in liver elastography studies (CT) |
| Shepp et al. [24] | Image reconstruction (MRI [b]) |
| Koay et al. [25] | Image reconstruction (MRI) |
| Collins et al. [26] | Filtering techniques validation (MRI) |

[a] Ultrasound; [b] magnetic resonance imaging.

This dataset and metrics mathematical functions allow the evaluation processes of performance techniques or methods that are immersed in the context of digital image processing. They are created with the purpose of computationally addressing the problem of the impact that the mentioned imperfections have on the quality of the information present in images associated with cardiac CT. In the third section of this article, we briefly describe the basic aspects related to the aforementioned evaluation process considering an article of our authorship (see Ref. [28]).

## 2. Data Description

In order to simulate the Poisson noise and both dark band and staircase artifacts present in the cardiac tomography bases, six synthetic DBs were generated and identified, as shown in Table 2.

**Table 2.** Synthetic DBs notation and identification of the imperfections considered.

| Database | Embedded Imperfections |
|---|---|
| Ground truth (DB1) | Ninguna |
| Poisson (DB2) | Poisson noise |
| Stair-step (DB3) | Stair-step artifact |
| Streak (DB4) | Streak artifact |
| Artifacts (DB5) | Both artifacts |
| Hybrid (DB6) | All (artifacts and Poisson noise) |

Considering a real cardiac CT database, images with information about both the left ventricular cavity (LVC) and the left ventricle wall (LVW) were selected. Then, the gray levels from these images were used to construct each of the aforementioned databases. This real database was acquired with a GE Bright Speed Elite 16-slice scanner and was donated by the Laboratory of Traitement signal et de l'image, belonging to Rennes I University, France.

Thus, to generate all synthetic DBs, specific gray levels were considered, and the myocardium and LV wall was simulated by an inner cone and an outer cone, respectively. The spatial resolution of both cones was $256 \times 256 \times 50$ voxels. The value 256 is equivalent to 50% of the spatial dimension of a typical cardiac tomography layer, whereas the value 50 corresponds to approximately 20% of the number of layers of the actual data analyzed.

All the databases were built in a set of concentric circular circles or layers, which were conditioned to contain densitometric information obtained by considering the average gray levels (GL) exhibited by a real cardiac computed tomography database (CT_DB). To

introduce in the databases the information related to the GLs, each of the generated circular layers was processed with a filling technique based on a region growing algorithm [29,30]. The following is a description of each of the DBs presented in Table 2.

### 2.1. DB1: Ground truth

To build this DB, 100 circular layers of variable radius were generated, with a step size of 1.5 pixels (px) and a common center located at the geometric center of each layer. The ordered grouping of these layers allowed the construction of the aforementioned cones. The first 50 circles, with radii taken from the close interval between 10 and 80.5 px, were grouped concentrically to generate an inner subcone, while the remaining 50 circles were grouped concentrically to generate an outer subcone whose radii were selected from the close interval between 50 and 120.5 px. The sum of these subcones generated the DB1, whose final physical structure resembled a truncated double cone, composed of the inner and outer subcones (see Table 3).

**Table 3.** Parameters of the subcones used to simulate LV characteristics.

| Subcone | Ranges for Radii (Smaller Radius: Larger Radius) | Circular Layers Intensity |
|---------|--------------------------------------------------|---------------------------|
| Inner   | 10:80.5                                          | 1500 [a]                  |
| Outer   | 50:120.5                                         | 1000 [b]                  |

[a] LVC gray level average; [b] LVW gray level average.

Figure 3 shows a volumetric view and three two-dimensional orthogonal views of images belonging to DB1.

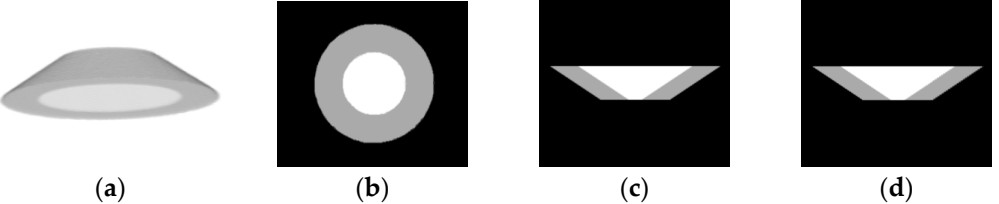

| (**a**) | (**b**) | (**c**) | (**d**) |

**Figure 3.** DB1 (ground truth): (**a**) volumetric view. (**b**) Axial view. (**c**) Coronal view. (**d**) Sagittal view.

### 2.2. DB2: Poisson

To generate this DB, DB1 was contaminated with Poisson noise using the algorithm proposed by Devroye [31]. Figure 4 presents a sequence of images belonging to DB2. More complete information about this database is offered in Section 3.1.

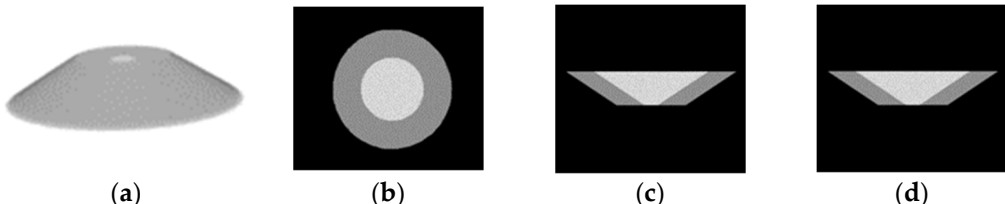

| (**a**) | (**b**) | (**c**) | (**d**) |

**Figure 4.** DB2 (Poisson): (**a**) volumetric view. (**b**) Axial view. (**c**) Coronal view. (**d**) Sagittal view.

### 2.3. DB3: Stair-Step

Knowing that the real database analyzed was acquired at 16 slices, the stair-step artifact is likely to occur in the vicinity of layers 16, 32 and 48. For this reason, they are the reference slices. In order to simulate the staircase, in both the inner and outer cone, the coordinates of the center of the layers were shifted by one pixel in the next group of slices:

(a)    14-15-17-18.
(b)    30-31-33-34
(c)    46-47-49-50

Figure 5 presents a volumetric view of the double cone representing this database and 2-D slices of DB3 images.

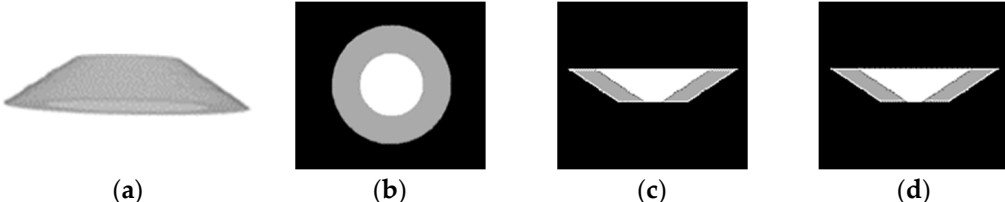

(**a**)                    (**b**)                    (**c**)                    (**d**)

**Figure 5.** DB3 (stair-step): (**a**) volumetric view. (**b**) Axial view. (**c**) Coronal view. (**d**) Sagittal view.

*2.4. DB4: Streak*

The streak artifact was simulated by decreasing the gray level of layers 11 and 41 in both cones by a value of 250 Hounsfield units (HU). This value was calculated considering 20% of the variation of the gray level in the layers of the analyzed base in which this artifact is present. The choice of the layers in which this artifact was simulated was made with the assumption that no positional coincidence with the staircase artifact occurred. 3-D view and a sequence of DB4 images are shown in Figure 6. The changes in gray levels are presented in Figure 7.

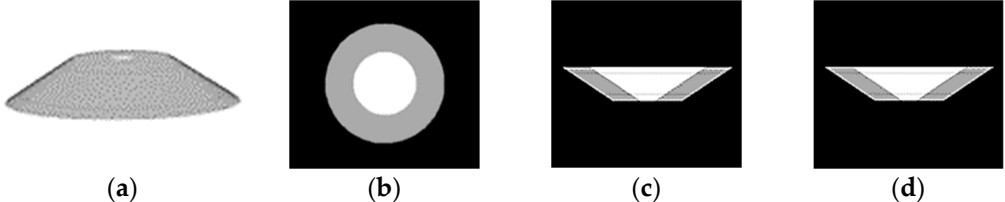

(**a**)                    (**b**)                    (**c**)                    (**d**)

**Figure 6.** DB4 (streak): (**a**) volumetric view. (**b**) Axial view. (**c**) Coronal view. (**d**) Sagittal view.

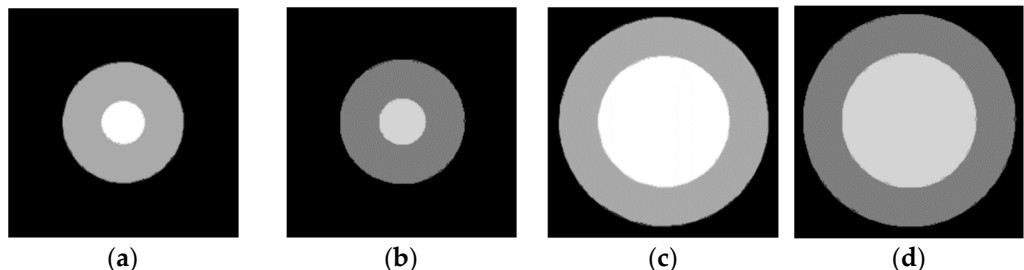

(**a**)                    (**b**)                    (**c**)                    (**d**)

**Figure 7.** Variation gray levels (GL) in DB4. (**a**) GL in slice 10. (**b**) Modified GL in slice 11. (**c**) Original GL in slice 40. (**d**) Modified GL in slice 41.

*2.5. DB5: Artifacts*

Following a similar process as described for the artifacts, a double cone was constructed by incorporating the band and ladder artifacts to the original double cone. Figure 8 is a sequence of 3-D and 2-D images of the aforementioned base.

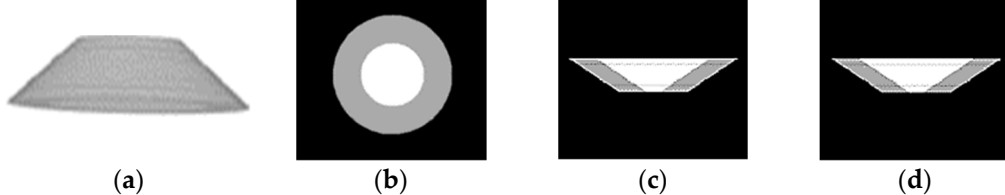

**Figure 8.** DB5 (artifacts): (**a**) volumetric view. (**b**) Axial view. (**c**) Coronal view. (**d**) Sagittal view.

### *2.6. DB6: Hybrid*

Applying similar processes to those described for both artifacts and Poisson noise, DB1 was used as a reference to construct a double cone incorporating first the Poisson noise and then the two artifacts already mentioned. Figure 9 represents a sequence of images of the referenced baseline.

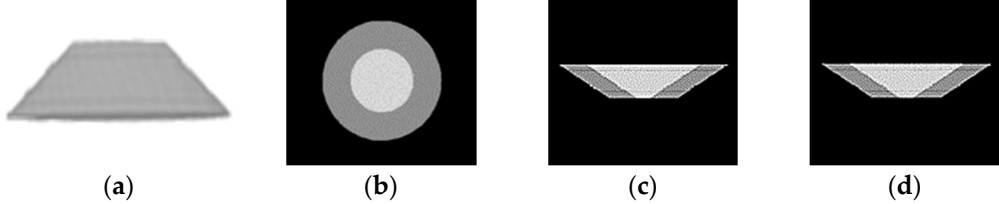

**Figure 9.** DB6 (hybrid): (**a**) volumetric view. (**b**) Axial view. (**c**) Coronal view. (**d**) Sagittal view.

## 3. Methods

### *3.1. Processes Involved in the Generation of the DB3 (Poisson) Database*

An analysis of the information presented in the previous section reveals that, with the exception of DB2, the generation of the other databases was based on procedures, with minimal levels of complexity, derived from the re-creation of the imperfections that affect the quality of MSCT images (noise and artifacts). In that sense, the essence of these procedures is summarized as follows:

1.  Direct assignment of averaged gray levels (DB1).
2.  Simulation of the staircase artifact by a minimal modification of the geometric center corresponding to layers located at key positions identified according to the actual tomographic data acquisition process (DB3).
3.  Variation of typical gray levels in specific, arbitrary layers (DB4)
4.  Integrated incorporation of information to generate, in the same database, more than one imperfection (DB5 and DB6).

The database that requires a special method for its generation is DB2, which is linked to Poisson noise. In order to generate DB2, a description of the method that allows to contaminate DB1 with this type of noise is presented [31–33].

According to Ahrens and Dieter [32], the mathematical model, presented in the summary using Equation (1), works adequately in a non-computational context limited to small $k$ values. A function developed in the MATLAB computational environment, called poissond.m, states that values of k no larger than 15 define the range of such adequate performance (see the MATLAB documentation for this function).

However, for computational applications, it is more convenient to replace Equation (1) with an iterative expression obtained by trying to generate a future probability value, $p(k+1)$, from a current one, $p(k)$, which is given by Equation (2):

$$p(X = k + 1) = \frac{\lambda}{k + 1} \cdot p(X = k), \text{ with } p(X = k) \neq 0 \tag{2}$$

For values of $k > 15$, the Stirling approximation [33] must be considered to generate the mathematical model represented by Equations (3) and (4):

$$p(X = k) = \frac{1}{\sqrt{2\pi k}} e^{[kln(1+v)-(\lambda-k)-\delta]} \tag{3}$$

where:

$$v = \frac{\lambda - k}{k} \; y \; \delta = \frac{1}{12k} - \frac{1}{360k^3} + \frac{1}{1260k^5} \tag{4}$$

If $v < 0.25$, the rounding errors of Equation (3) are notable; therefore, the argument of the exponential function ($e$) must be expanded, excluding $\delta$, obtaining Equations (5) and (6):

$$p(X = k) = \frac{1}{\sqrt{2\pi k}} e^{[kv^3\phi(v)-\delta]} \tag{5}$$

where:

$$\phi(v) = \frac{1}{2} + \frac{v}{3} - \frac{v^2}{4} + \frac{v^3}{5} - \ldots \tag{6}$$

In summary, if $k \leq 15$, it is necessary to use Equation (2) in order to contaminate DB1 with Poisson noise and generate the synthetic database DB3; otherwise, Equations (5) and (6) are considered. The changes produced in DB1 are shown in Figure 10.

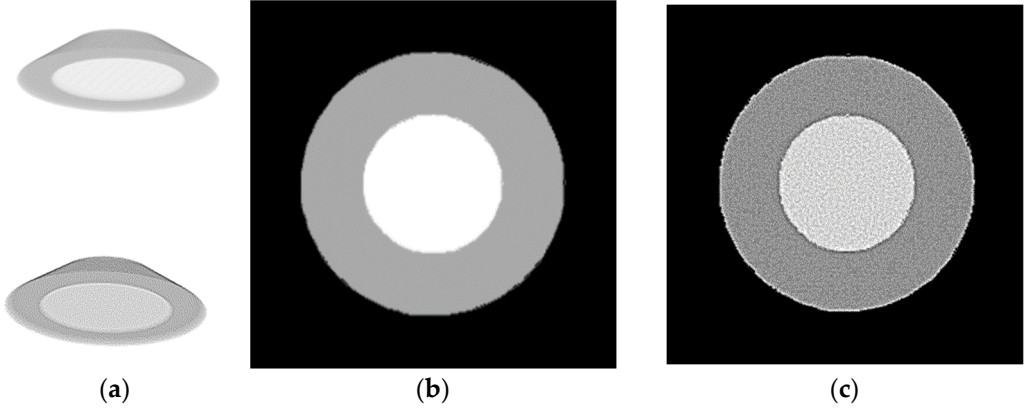

(a)　　　　　　　　　　　　(b)　　　　　　　　　　　　(c)

**Figure 10.** Graphical comparison between DB1 and DB2. (**a**) volumetric view (top: DB1 and bottom: DB2). (**b**) DB1 (ground truth) 2-D view. (**c**) DB2 (Poisson) 2-D view.

Lastly, due to the fact that DB3 will be corrupted with noise, the metric called peak signal-to-noise ratio (PSNR) was calculated using DB1 and DB3 images [34]. The value obtained for PSNR was 39.02 decibels.

*3.2. Possible Scenarios for the Use of the Described Phantoms*

In Ref. [28], the use of a digital image processing technique is reported based on the similarity enhancement of the information contained into a real cardiac DB, the DB1 and DB6 databases described in Sections 2.1 and 2.6, respectively.

In sum, the objective in Ref. [28] was to propose a global index or function score, based on both blind and full-reference metrics, for the determination of the quality of the similarity enhancement process of the main features present in the aforementioned images. The use of ground truth (DB1) and its filtered versions with the similarity enhancement method allow development of the processes established for the reference metrics, such as, for example, PSNR. The results obtained for DB2 correspond to the findings generated by applying the aforementioned enhancement to the real DBs, thus demonstrating the true usefulness of the numerical phantoms described in this article. Therefore, researchers are encouraged to use this synthetic dataset to validate digital image processing techniques related to cardiac computed tomography.

Finally, it is important to note that the similarity enhancement method represents a valuable digital image preprocessing tool based on the similarity criterion proposed by Haralick and Shapiro [35] and, in a general way, has been considered to raise the quality of information associated with medical images [36]. The distinctive feature of this technique is that it allows relating a highly contrasted version of an image to another smoothed version of the original image, and such versions can be linked to images processed with high-pass and low-pass filters, respectively. It can be asserted that this method has been widely used for the enhancement of multilayer computed tomography images in both two-dimensional and three-dimensional space, the most representative references being those reported in Refs. [37,38].

The quality of synthetic images or phantoms can be evaluated considering metrics designed for this purpose. In Ref. [39], a score function is built considering a set of metrics functions. In the present article, only with the purpose of illustrating how to measure the effect of digital filtering when processing a synthetic database, we consider the PSNR. The results obtained after applying the similarity enhancement method, Gaussian and anisotropic diffusion filters to DB6 are presented in Figure 11 [40,41]. The values obtained for the signal-to-noise ratio peak are shown in Table 4. The best result was obtained using a diffusion filter.

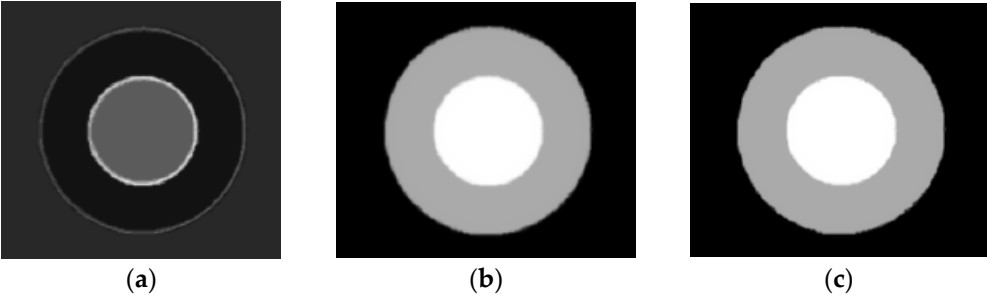

| (a) | (b) | (c) |

**Figure 11.** DB6 processed with: (**a**) similarity method. (**b**) Gausian filter. (**c**) Anisotropic diffusion filter.

**Table 4.** Values of PSNR considering 3 different digital processing imaging techniques.

| Processing Imaging Technique | PSNR * (Decibels) |
| --- | --- |
| Similarity method | 39.23 |
| Gaussian filter | 39.37 |
| Anisotropic diffusion filter | 39.60 |

* Reference PSNR = 39.02 decibels.

In conclusion, a comprehensive description and use of synthetic databases useful in cardiac CT imaging processing is performed. These DB have features considered during the tuning process related to the performance of new image processing techniques, which usually require ground truth and recreation of typical imperfections present in real CT datasets.

**Author Contributions:** Conceptualization, M.V., A.B. and R.M.; methodology, M.V.; software, M.V.; validation, M.V. and A.B.; formal analysis, M.V. and A.B.; investigation, M.V. and A.B.; resources, M.V.; data curation, M.V.; writing—original draft preparation, M.V.; writing—review and editing, M.V., A.B. and R.M.; visualization, M.V.; supervision, M.V. and R.M.; project administration, M.V.; funding acquisition, M.V. All authors have read and agreed to the published version of the manuscript.

**Funding:** This research received no external funding.

**Institutional Review Board Statement:** Not applicable.

**Informed Consent Statement:** Not applicable.

**Data Availability Statement:** https://dx.doi.org/10.21227/tjr6-tf41 (accessed on 15 June 2022).

**Conflicts of Interest:** The authors declare no conflict of interest.

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
