# Peer review of "Description and Use of Three-Dimensional Numerical Phantoms of Cardiac Computed Tomography Images"

_data, 2022_

Round 1

Reviewer 1 Report

It's well described enough to know your efforts. I wrote my comment in PDF file using review-note function of Adobe Reader. 

Author Response

To answer the referee's comments and improve the quality of the article, we incorporate the following actions:

1. Do you mention general CT which are used in hospital for patients? 
All CT are scanned multilayer, serial images, and so on. 
Do MSCT have anything special, unlike general CT?

Only the term CT was considered and the MSCT was eliminated from the discourse. 
In addition, the following paragraph was added to clarify this situation:

"Although there are several sub-modalities within computed tomography, the differences are not relevant. 
In this article, CT is considered the modality that genuinely represents the main characteristics of computed images 
generated by X-ray emission."

2. Basically, why you use CT with artifacts? 
There are artifact-less CT in relatively big hospitals.

It is a known fact that artifact-free CT scanners are available in relatively large hospitals. 
However, as is indicated in section two, the real CT dataset available at the time of the creation of the synthetic database was acquired with a GE Bright Speed Elite 16-slice scanner, which introduces the aforementioned artifacts.

3. Write exactly the direction. a - sagittal; b - coronal ?
These changes were incorporated in figure 1

4. They are not bottom, middle, and top positions.

The words: bottom, middle, and top were changed using the words: left, center, and right (see figure 2).

In addition, the correction of the English language and style was applied in several sections of the article.

Reviewer 2 Report

Authors present a database including numerical phantoms representing cardac CT images. 

The work present soundness in the method and in the content and it is interesting. 

On the abstract is not clear the concept of "deterioration of the information" in the image due to the imperfections. The word deterioration  refers  to changes in the image during the time, which does not seem to be the case in digital images. Please clarify the idea. 

I recommend to the authors to include more examples of phantoms to visualize better the results.

Are there a way to evaluate the "quality" of the phantoms ? Authors may discuss that and relate it to the possible scenarios of use of the phantoms that they describe in section 3.2.

A conclusion must be added. 
